# Medicine-Food Herbs against Alzheimer’s Disease: A Review of Their Traditional Functional Features, Substance Basis, Clinical Practices and Mechanisms of Action

**DOI:** 10.3390/molecules27030901

**Published:** 2022-01-28

**Authors:** Pengfei Guo, Baoyue Zhang, Jun Zhao, Chao Wang, Zhe Wang, Ailin Liu, Guanhua Du

**Affiliations:** 1State Key Laboratory of Bioactive Substances and Functions of Natural Medicines, Institute of Materia Medica, Chinese Academy of Medical Sciences and Peking Union Medical College, Beijing 100050, China; guopf@imm.ac.cn (P.G.); zhangbaoyue@imm.ac.cn (B.Z.); zhaojun@imm.ac.cn (J.Z.); wangchao@imm.ac.cn (C.W.); wangzh@imm.ac.cn (Z.W.); 2Beijing Key Laboratory of Drug Target Identification and Drug Screening, Institute of Materia Medica, Chinese Academy of Medical Sciences and Peking Union Medical College, Beijing 100050, China

**Keywords:** Alzheimer’s disease, traditional Chinese medicines, medicine food homology, broad-spectrum activities, drug development

## Abstract

Alzheimer’s disease (AD) is a progressive, neurodegenerative disorder that currently has reached epidemic proportions among elderly populations around the world. In China, available traditional Chinese medicines (TCMs) that organically combine functional foods with medicinal values are named “Medicine Food Homology (MFH)”. In this review, we focused on MFH varieties for their traditional functional features, substance bases, clinical uses, and mechanisms of action (MOAs) for AD prevention and treatment. We consider the antiAD active constituents from MFH species, their effects on in vitro/in vivo AD models, and their drug targets and signal pathways by summing up the literature via a systematic electronic search (SciFinder, PubMed, and Web of Science). In this paper, several MFH plant sources are discussed in detail from in vitro/in vivo models and methods, to MOAs. We found that most of the MFH varieties exert neuroprotective effects and ameliorate cognitive impairments by inhibiting neuropathological signs (Aβ-induced toxicity, amyloid precursor protein, and phosphorylated Tau immunoreactivity), including anti-inflammation, antioxidative stress, antiautophagy, and antiapoptosis, etc. Indeed, some MFH substances and their related phytochemicals have a broad spectrum of activities, so they are superior to simple single-target drugs in treating chronic diseases. This review can provide significant guidance for people’s healthy lifestyles and drug development for AD prevention and treatment.

## 1. Introduction

Alzheimer’s disease (AD) is a specific onset and process of cognitive and functional decline with particular neurological characteristics, especially in people over 65 years of age with high morbidity and mortality [1,2]. Prince M et al. estimated that 65.7 million people in 2030 and 115.4 million people in 2050 will live with dementia worldwide, which implies an extremely serious burden to global public health and social welfare [3]. Deficits in the ability to encode and store new memories characterize the initial stages of the disease. Subsequent progressive changes in cognition and behavior accompany the later stages. Due to the complexity of the etiology of AD, its pathological and physiological mechanisms have been controversial. To date, the core pathological hallmarks recognized by researchers are amyloid plaques, neurofibrillary tangles, synapses, and/or neuronal loss [4,5].

At present, the main licensed pharmacologic therapy of patients is cholinesterase inhibitors (donepezil, rivastigmine, and galantamine) and noncompetitive N-methyl-D-aspartate receptor (NMDA) antagonists (memantine). Cholinesterase inhibitors are licensed therapy for patients with mild-to-moderate Alzheimer’s disease, and NMDA antagonists are licensed therapy for patients with moderate-to-severe AD [6]. Other treatments include immunotherapy (bapineuzumab), amyloid aggregators (tramiprosate), tau aggregation inhibitors (methylthioninium chloride), glycogen synthase kinase 3 (GSK3) inhibitors (lithium), and natural products and vitamins (ginkgo biloba, omega 3 fatty acids, docosahexaenoic acid, and vitamin E) [6,7].

In recent years, with the improvement of public health awareness, people have paid increasing attention to the prevention and treatment of chronic diseases through diet, such as a “low-, fermentable oligo-, and di-, monosaccharides and polyols” diet, Mediterranean diet, a low-lactose diet, and medicinal food herb diet [8,9,10,11]. The concept of medicine food homology (MFH) originally came from the (Huangdi’s) Internal Classic in ancient China, which means that Chinese medicine and food are of the same origin, and, thus, they can be used as food and administered to patients as medication to regulate the human body’s metabolic disturbances or imbalances [11,12].

In China, many traditional Chinese medicines (TCMs) are functional foods and dietary supplements, such as *Panax ginseng C.A. Mey.*, *Crocus sativus* L., and *Angelica acutiloba* (*Siebold & Zucc.*) *Kitag*. For cultural reasons, herbal remedies are called alternative therapies in North America and other countries [8]. MFH substances with abundant resources have exhibited pharmacological effects at preventing and treating a variety of diseases, including cancers, diabetes, cardiovascular diseases, Alzheimer’s disease, Parkinson’s disease, viruses, influenza, and, in particular, chronic diseases [13]. To ensure the safety of functional foods from TCM, the National Health Commission of the People’s Republic of China released specific provisions on MFH items, which considered 109 TCMs as functional foods for alleviating or curing various chronic diseases through the diet as of 2019 [8].

For this paper, the literature review covers the period between 2001 and 2021 relating to MFH items with antiAD effects. The literature was collected from the SciFinder, EMBASE, Mendeley, Google Scholar, PubMed, and Web of Science databases. The keywords used for the literature search were as follows: “Medicine-food Herb and Alzheimer’s disease”, “Medicine-food Herb and chemical components or phytoconstituents or natural products”, and “Medicine-food Herb and clinical evidence and Alzheimer’s disease”. Then, we tried to classify all data pertaining to the pharmacological effects of MFH functional foods on memory in different animal models, methods, mechanisms, etc. The summarized results comprised a total of 28 plant species, 35 phytochemicals, and 9 TCM formulations of MFH resources with antiAD activities exhibited in various preclinical and five clinical studies (Table 1, Table 2, Table 3, Table 4 and Table 5). The main MFH items that will be introduced are those, which are often used, such as “Monarch medicine” and clinical evidence, including *Angelica acutiloba* (*Siebold & Zucc.*) *Kitag.*, *Panax ginseng C.A. Mey.*, and *Poriacocos* (*Schw.*) *Wolf*. This review will provide further support for the basic and clinical research on MFH and the use of functional foods in a healthy lifestyle in the future.

## 2. Medicine and Food Herb Extract and Its Ingredients against Alzheimer’s Disease

### 2.1. Panax Ginseng C.A. Mey

#### 2.1.1. Traditional Functional Features

Panax ginseng, an herb from the Araliaceae family, is a qi-tonifying medicinal substance with the effects of tranquilizing the mind and replenishing wisdom [129]. Its supposed health benefits are such that it is used in many countries. Ginseng is “generally recognized as safe” (GRAS) for consumption by adults by the U.S. Food and Drug Administration [130].

#### 2.1.2. MOAs of the Extract

Korean white ginseng, red ginseng extract, and black ginseng are the main processed products of ginseng. Lee et al. reported that their extracts can inhibit AChE and BuChE in a concentration-dependent manner in scopolamine (SCOP)-induced amnesic mice. The IC50 values are 1.72 mg/mL, 5.89 mg/mL, and 6.30 mg/mL [16]. Simultaneously, the processed products of ginseng showed their antiAD mechanisms. Specifically, in vitro and in vivo findings indicated that red ginseng significantly ameliorated AD-related pathologies, such as Aβ deposition, gliosis, and neuronal loss [33]. Fermented ginseng extract enhanced behavioral memory function in an effort to reduce Aβ levels in ICR mice and TG mice and in HeLa cells [34]. Furthermore, white ginseng extract (100 and 500 mg/kg/day) could improve memory impairment by attenuating neuronal damage and neuroinflammation caused by AβO in intrahippocampal AβO (10 μM)-injected mice [35] (Figure 1).

In addition, in vivo and in vitro pharmacological studies have shown that ginseng plays a key role in antiAD as a “Jun medicine” in TCM prescriptions, such as Shenmayizhi decoction (SMYZD), Yizhi Qingxin formula (YZQXF), and Jangwonhwan.

Evidence has revealed that YZQXF alleviates cognitive deficits by increasing the levels of LC3II/LC3I and Beclin1, remarkably activating 4EBP1 and inhibiting the phosphorylation of mammalian target of rapamycin (mTOR) at Ser2448 in early-onset AD mice [24]. Researchers also showed that YZQXF ameliorated cognitive decline by decreasing BDNF, TrkB, TNFα, IL-2, and IL-6 levels and upregulating acetylcholine and IL-10, thus promoting the activation of Erk and Akt signaling in APP/PS1 mice [23]. In an in vivo model of AD, SMYZD at concentrations of 11.88 g/kg/d can significantly improve memory abilities associated with an increase in the levels of NG2 and Ang1 protein expression levels in Wistar rats [22]. An in vitro study showed that Jangwonhwan at 400 mg/kg/day notably reduced Abeta (1–42) levels and beta-amyloid deposition and partially suppressed oxidative stress accumulation in SH-SY5Y neuroblastoma cells [25].

#### 2.1.3. MOAs of Ingredients

Ginsenoside Rg1, ginsenoside Rh2, and ginsenoside Rd are the main saponins of Panax ginseng. Evidence has indicated that ginsenoside Rg1 can improve cognitive ability, protect NSCs/NPCs, and promote neurogenesis by enhancing antioxidant (GSH-Px, SOD) and anti-inflammatory (IL-1β, IL-6, and TNF-α) capacities in the hippocampus [75]. Ginsenoside Rh2, another saponin component of ginseng, was reported to improve learning and memory function by inhibiting cholesterol and lipid raft concentrations, thus reducing amyloid-beta secretion and APP endocytosis in tg2576 mice [76]. Additionally, ginsenoside Rd (10 mg/kg for 7 days) prevented okadaic acid-induced neurotoxicity through the upregulation of PP-2A in cultured cortical neurons (2.5 or 5 μmol/L for 12 h) [77] (Figure 2).

Notably, gintonin, a novel lysophosphatidic acid–protein complex in ginseng, can exert nervous system protection through the LPA receptor-Gαq/11 protein-phospholipase C-IP3 receptor-[Ca^2+^] I transient pathway [78,131]. In addition, ginsenoside Compound K is a metabolite of panaxadiol generated by the metabolic actions of intestinal flora in humans. A study demonstrated that Compound K promoted the clearance of Aβ by enhancing autophagy via the mTOR signaling pathway in primary astrocytes [79].

#### 2.1.4. Clinical Evidence

In an open-label study, the Mini-Mental State Examination (MMSE) and AD Assessment Scale (ADAS) began to show improvements and continued up to 12 weeks (*p* = 0.029 and *p* = 0.009 vs. baseline, respectively) during 12 weeks of ginseng treatment. The results indicated that Panax ginseng is clinically effective in the treatment of AD. Simultaneously, three randomized open-label studies have shown that Korean Red Ginseng extract is clinically effective and gradually improves cognitive function in AD patients [14,15,16].

### 2.2. Crocus sativus L.

#### 2.2.1. Traditional Functional Features

*Crocus sativus* L., a stemless herb that belongs to the Iridaceae family, has blood-activating and menstruation-regulating properties and has been considered an edible food since ancient times. It is mainly used in activating blood circulation and eliminating blood stasis, removing heat from blood and counteracting toxicity, and calming the nerves [132].

#### 2.2.2. MOAs of the Extract

In recent years, researchers have found that *Crocus sativus* L. has various useful pharmacological properties, such as anticonvulsant, antidepressant, anti-inflammatory, antitumor, and learning and memory-improving effects [133]. A review by Hatziagapiou K et al. summarized the anti-AD mechanism of *Crocus sativus* L., including the following: antioxidative stress, inhibitory effects of Aβ fibrillogenesis, inhibitory effects of tau aggregation, reduction of glutamatergic synaptic transmission, restoration of cellular antioxidant defenses, and inhibition of cellular apoptosis or death [134]. For example, Soeda et al. designed an experiment to investigate the neuroprotection mechanisms of *Crocus sativus* L. by inhibiting cellular apoptosis/death. It was found that *Crocus sativus* L. prevented ethanol-induced impairment of learning and memory with a marked reduction in the caspase-3 and TNF-alpha-induced release of cytochrome c from the mitochondria in PC-12 cells [36] (Figure 1).

#### 2.2.3. MOAs of Ingredients

The in vitro and in vivo research of Ochiai T et al. showed that crocin, a constituent of *Crocus sativus* L., effectively promoted mRNA expression of gamma-glutamylcysteinyl synthase, contributing to GSH synthesis in mice and, thus, inhibiting neutral sphingomyelinase activity and ceramide formation [135]. Another study indicated that *Crocus sativus* compounds (trans-Crocin 4 and trans-Crocetin) suppressed the levels of GSK3β and ERK1/2 kinases and significantly reduced total tau and tau phosphorylation, which may be promising candidates in the prevention and treatment of AD [80] (Figure 2).

#### 2.2.4. Clinical Evidence

In a double-blind and randomized study on patients with mild-to-moderate AD, 54 Persian-speaking adults were randomly assigned to receive a capsule of saffron at 30 mg/day (15 mg twice per day) or donepezil at 10 mg/day (5 mg twice per day). This phase II study (Clinical Trials Registry: IRCT138711051556N1) provides preliminary evidence that saffron extract may be used as a potential TCM for mild-to-moderate AD patients [17]. A 16-week, randomized and placebo-controlled study on patients randomly assigned to receive 30 mg/day of saffron capsule or placebo capsule (two capsules/day) demonstrated comparable results with the placebo capsule, and saffron was both safe and effective in mild-to-moderate AD in the short-term [18]. Another single-blind randomized clinical trial by the Tsolaki M et al. group also showed that Crocus is efficacious and safe for the management of amnesic and multidomain MCI [19].

### 2.3. Cistanche afghanica Gilli

#### 2.3.1. Traditional Functional Features

As a tonic herb, Herba Cistanches has earned the honor of “Ginseng in the deserts” with its function of reinforcing the kidney, replenishing vital essence and blood, and inducing taxation [136].

#### 2.3.2. MOAs of the Extract

Herba Cistanches, a yang-tonifying medicine in China, possesses broad medicinal functions in immunomodulation, endocrine regulation, hepatoprotection, and antiaging, antibacterial, antiviral, and antitumor [137]. In recent years, several excellent reviews have described that Herba Cistanches have great potential in the treatment of age-related diseases. On the one hand, the mechanisms were mainly involved in blocking Aβ 1–42 amyloid deposition, decreasing P-tau phosphorylation, inhibiting the formation of NO, affecting the signaling pathway between ROS, and the opening of Ca^2+^ channels. On the other hand, Herba Cistanches could induce neuroprotection by inducing cell cycle arrest and apoptosis, increasing the expression of the antiapoptotic proteins, stimulating intestinal epithelial cell proliferation, upregulating TGF-β and CRMP-2 expression levels, and improving antioxidant enzymes [138,139,140].

#### 2.3.3. MOAs of Ingredients

It was also found that phenylethanoid glycosides from *Cistanches salsa* (10, 50 mg/kg) were able to protect dopaminergic neurons against dopamine neurotoxicity induced by 1-methyl-4-phenyl-1,2,3,6-tetrahydropyridine in C57 mice [141]. Another interesting finding indicated that echinacoside, a phenylethanoid from the stems of *Cistanche salsa*, showed both antiapoptotic and anti-inflammatory properties. Pretreatment with echinacoside also significantly inhibited inflammatory mediators, such as TNF-α, IL-1β, and IL-6 [82] (Figure 2).

In addition, tubuloside B, a phenylethanoid of the stems of *Cistanche salsa*, has neuroprotective effects, such as antioxidative stress and antiapoptotic actions (Figure 3). Deng et al. then revealed that tubuloside B (1, 10, or 100 mg/L) attenuated TNF-a-induced apoptosis and the accumulation of intracellular ROS and [Ca^2+^]I in SH-SY5Y neuronal cells [81].

#### 2.3.4. Clinical Evidence

Li N, et al. evaluated the neuroprotective effects of Cistanches Herba on patients with moderate AD. Conclusions were obtained in their studies that Cistanches Herba had potential neuroprotective effects for moderate AD by reducing the levels of T-tau, TNF-α, and IL-1β [20].

### 2.4. Angelica acutiloba (Siebold & Zucc.) Kitag.

#### 2.4.1. Traditional Functional Features

*Angelica acutiloba* (*Siebold & Zucc.*) *Kitag.* (RAK), an ancient Chinese herb, can be found in Divine Farmers Materia Medica (Han Dynasty, A.D. 25–220). It is a blood-tonifying medicine that can activate blood circulation, regulate menstruation, relieve pain and feminine hernias, and is also called “female ginseng” [142].

#### 2.4.2. MOAs of the Extract

RAK (Danggui), as a main medicine, plays a major role in many TCM tonic blood prescriptions, such as Danggui Buxue Tang (DGBXT), Danggui-Shaoyao-San (DGSYS), and Fo Shou San (FSS).

DGBXT, a historical Chinese herbal decoction, contains just two MFH materials: *Astragali Radix* and *Radix Astragali*. An in vitro assay demonstrated that DGBXT protected β-amyloid-induced cell death by altering the expression ratio of Bcl2 to Bax and markedly suppressed the Aβ-induced expression of apoptotic markers (cleaved-caspase 3/9 and PARP) in cultured cortical neurons [26]. Emerging evidence also showed that DGSYS could increase E2, NO, and glycine in SAMP8 mice, contributing to cognitive improvement [27]. Another study showed that DGSYS ameliorated oxidative stress and inflammation by decreasing the levels of PEG2, TXB2, and LTB4 and inhibiting the expression of cPLA2, COX-1, and COX-2, thus ameliorating cognitive deficits in APP/PS1 mice [28]. The FSS formula was reported to ameliorate the symptoms of AD by regulating the gut–liver–brain axis, oxidative stress (MDA), and gastrointestinal tract barrier in APP/PS1 mice [29].

#### 2.4.3. MOAs of Ingredients

RAK and *Astragalus aaronii* (*Eig*) *Zohary* are both important MFH substancess, and many of their active ingredients have been tested in vivo and in vitro to prove their therapeutic effect on AD, such as decursinol, vanillic acid, and ligustilide [143,144]. Pretreatment of mice with decursinol significantly attenuated the Abeta (1–42)-induced impairment in passive avoidance performance in ICR mice [84]. Additionally, decursinol exerted a neuroprotective effect against Aβ25–35-induced neurotoxicity in PC12 cells by suppressing the mitochondrial pathway of cellular apoptosis [85]. In addition, a study showed that vanillic acid reduced AChE, TNF-α, and corticosterone, thus improving antioxidants to contribute to neuroprotection in Swiss albino mice [83].

Importantly, Z-ligustilide (40 mg/kg) significantly ameliorated both AD-related neuropathological signs (Abeta, amyloid precursor protein, and phosphorylated Tau immunoreactivity) and proinflammatory mediators (TNF-alpha and NF-kappaB) in the hippocampus of Wistar rats [86]. In addition, Z-ligustilide can resist the neurotoxicity induced by β -amyloid by simultaneously regulating p38 and PI3-K/Akt pathways in SH-SY5Y cells and PC12 cells [87]. It is worth noting that ligustilide has been demonstrated to permeate the blood–brain barrier in freely moving rat brains, suggesting that ligustilide may be a promising compound for the treatment of AD [145] (Figure 3).

### 2.5. Astragalus aaronii (Eig) Zohary

#### 2.5.1. Traditional Functional Features

*Astragalus aaronii* (*Eig*) *Zohary*, derived from the roots of Astragalus membranaceus and A. membranaceus var. mongholicus, is a qi-tonifying herb that is widely used in China and shows potent cardiovascular protective effects [146].

#### 2.5.2. MOAs of the Extract

*Astragalus aaronii* (*Eig*) *Zohary* plays a powerful role in the Tiaoxin Recipe and Fuzheng Quxie Decoction (FZQXD). Compared with the normal control group, treatment with the Tiaoxin Recipe decreased the content of serum Aβ1–42 accumulation in APP/PS1 mouse serum. Studies have also shown that microRNA-34a (miR-34a) expression can significantly lead to neuronal apoptosis [147]. The administration of a Tiaoxin Recipe significantly decreased miR-34a expression in APP/PS1 mice [30]. In vivo experiments confirmed that FZQXD could exert a neuroprotective effect on AD via VEGF and VEGF-receptor signaling [31].

#### 2.5.3. MOAs of Ingredients

Astragaloside IV (AS-IV) is a small molecular weight (MW = 784 Da) saponin purified from Astragalus membranaceus that has shown multiple beneficial effects in the prevention and treatment of neurodegenerative disorders [148] (Figure 4). The literature has demonstrated that AS-IV prevents AβO-induced memory impairment by promoting the PPARγ/BDNF signaling pathway [88]. An in vivo study showed that AS-IV treatment increased PPARγ and BACE1 expression and reduced neuritic plaque formation and Aβ levels in the brains of APP/PS1 mice and ultimately attenuated the generation of Aβ [89]. In addition, pretreatment with AS-IV prevented Aβ1–42-induced SK-N-SH cell apoptosis by inhibiting the mPTP opening and ROS generation for the prevention and treatment of neurodegenerative disorders such as AD [90].

### 2.6. Poriacocos (Schw.) Wolf

#### 2.6.1. Traditional Functional Features

*Poriacocos* (*Schw.*) *Wolf*, an edible mushroom, grows underground on the roots of pine trees. It is a water-draining and swelling-dispersing herb for the treatment of edema, nephrosis, gastric atony, acute gastroenteric catarrh, dizziness, and nausea. *Poria cocos* is often commercially available and is popularly used in Asia [149].

#### 2.6.2. MOAs of the Extract

Extracts from *Poriacocos* (*Schw.*) *Wolf* have various pharmacological properties, including anti-inflammatory, immunomodulatory, anticancer, and antihyperglycemic effects [150]. Huang et al. found that *P. cocos* water extract significantly attenuated the UCMS-induced turnover rate of DA and 5-HT and inhibited the UCMS-induced activated inflammatory response (p38, NF-κB, and TNF-α) in the frontal cortex by using an SD rat model [38]. Sun Y et al. applied the APP/PS1 mouse model to study the effects of learning and memory and the mechanisms of the aerial part of *Poria cocos*. The results showed that this medicinal plant, which is also a functional food, might ameliorate cognitive function by reducing the formation of Aβ, increasing its clearance and reversing gut microbiota dysbiosis [37] (Figure 1).

In addition, *Poriacocos* (*Schw.*) *Wolf* is the most important herb of Lingui Zhugan decoction (LGZGD) with neuroprotective effects. LGZGD, a notable prescription for the treatment of AD, significantly improved learning and memory ability by regulating Aβ transportation and inhibiting RAGE/MAPK and NF-κB signaling [32].

### 2.7. Alpinia oxyphylla Miq.

#### 2.7.1. Traditional Functional Features

*Alpinia oxyphylla Miq.* (AOM), derived from the mature and dried fruits of the Zingiberaceae plant *Alpinia oxyphylla Miq.*, has good effects as a medicine and food in China. AOM is an astringent herb that arrests discharge due to insecure healthy qi and decreases visceral functions, such as excessive sweating, chronic diarrhea, and enuresis [151].

#### 2.7.2. MOAs of the Extract

Recently, a systematic overview of progress on the anti-AD mechanism of AOM was reported. Research has demonstrated that AOF exerts great therapeutic potential in the treatment of AD by inhibiting the formation of Aβ and phosphorylation of tau, increasing antioxidant capacity, anti-inflammatory and anti-apoptosis effects, and inhibiting the activity of acetylcholinesterase [41,152]. 

Wang et al. found that AOM extracts attenuated LPS-induced learning and memory impairment due to their inhibitory effect on neuroinflammation, amyloid -β (Aβ) deposition, and p-tau [40]. Chang et al. also confirmed in vivo and in vitro that AOM increased the expression of uPA, tPA, MMP-9, and MAPKs and induced ERK1/2, JNK, and p38 phosphorylation to exert nerve regeneration [39]. In addition, a study also showed that treating neurons with AOM extract (80–200 μg/mL) significantly increased cell viability and reduced the number of apoptotic cells, thus exerting neuroprotection [42]. 81Moreover, the butanol extract of AOM (180 mg/kg, 360 mg/kg) was reported to be efficacious in preventing neuronal damage by inhibiting β-secretase and the level of Aβ (1–42) in ICR mice [92] (Figure 1).

#### 2.7.3. MOAs of Ingredients

5-Hydroxymethylfurfural (5-HMF) and tetochrysin (TEC) are the main effective compounds of AOF ethanol extract and exert neuroprotective activities against AD (Figure 2 and Figure 3). For example, both 5-HMF and TEC significantly decreased the content of Aβ 1–42 and increased the activities of antioxidative enzymes, including SOD and GPx [91,153].

### 2.8. Zingiber officinale Roscoe

#### 2.8.1. Traditional Functional Features

*Zingiber officinale Roscoe* (ZOR) is the fresh rhizome of *Zingiber officinale Roscoe* traced from Shen Nong Ben Cao Jing. In the Chinese Pharmacopoeia, ZOR has the function of inducing perspiration and dispelling cold, warming the stomach, stopping vomiting, resolving phlegm, and relieving coughs [154].

#### 2.8.2. MOAs of the Extract

ZOR is a traditional pungent-cool exterior-releasing medicine and is widely used as an antioxidant, anti-inflammatory, antidiabetic, antinausea, neuroprotective, and cardiovascular protective agent [155].

Accumulated investigations have demonstrated that ginger positively affects memory function by anti-inflammatory effects in the treatment of AD. The results from the SD rat model revealed that the operation + high dose (4 g/kg) groups of ginger root extract had lower levels of the nuclear factor-κB (NF-κB) and interleukin-1β (IL-1β) expression than the operation + moderate dose (2 g/kg) and operation + low dose groups (1 g/kg) [43]. Further experiments in the hippocampus of Wistar rats revealed that ZOR extracts at a dose of 200 mg/kg significantly suppressed the inflammatory response of GFAP and IL-1β expression [44].

Mathew M et al. evaluated the antioxidant activity and cholinesterase inhibition properties of a methanolic extract of dry ginger (GE). The extract of total phenolic content and dry material contained 18 ± 0.6 mg/g gallic acid equivalents and 4.18 ± 0.69 mg quercetin equivalents/g, respectively. They found that GE expressed high antioxidant activity with an IC50 value of 70 ± 0.304 microg/mL in the DPPH assay and 845.4 ± 56.62 microM Fe (II) equivalents/g dry weight in the FRAP assay. They also found that GE had IC50 values of 41 ± 1.2 microg/mL and 52 ± 2 microg/mL for the inhibition of acetylcholinesterase and butyrylcholinesterase, respectively, in Ellman’s assay [46]. Another study found that water-extractable red and white ginger inhibited acetylcholinesterase (AChE) activities in a dose-dependent manner, and white ginger had higher acetylcholinesterase inhibitory activity than red ginger. Both extracts significantly decreased the MDA content induced by sodium nitroprusside and quinolinic acid in a dose-dependent manner [45] (Figure 1).

#### 2.8.3. MOAs of Ingredients

6-Gingerol, a phenolic compound, is the major pharmacologically active component in ginger rhizomes with diverse pharmacological activities, such as antitumor, anti-inflammatory, and antioxidant effects, etc. [156]. At present, there have been sufficient in vivo and in vitro experiments to prove that 6-gingerol has potential neuroprotective effects.

An in vitro experiment resulted in 6-gingerol exhibiting neuroprotective effects, and pretreatment with 6-gingerol significantly increased cell viability and reduced cell apoptosis in Aβ1–42-treated PC12 cells. Additionally, 6-gingerol pretreatment markedly reduced the levels of intracellular ROS, MDA, NO, and lactate dehydrogenase (LDH) leakage and increased SOD activity compared with the Aβ1–42 treatment group. Moreover, pretreatment with 6-gingerol (80, 120 μM) also markedly enhanced the protein levels of p-Akt and p-GSK-3β caused by Aβ1–42 [93].

In vivo, in a rat C57BL/6N mouse model induced by a high-fat, high-sugar Western diet (HFSD), 6-shogaol dose-dependently ameliorated obesity and emotional memory deficits [94]. In another study with an ICR mouse model, administration of 6-shogaol at a concentration of 10 mg/kg significantly elevated the expression levels of NGF, PSD-95, and synaptophysin in hippocampal tissues, compared with the vehicle-treated group [95].

### 2.9. Polygonatumodoratum (Mill.) Druce

#### 2.9.1. Traditional Functional Features

*Polygonatumodoratum* (*Mill.*) *Druce* (PMD), a kind of native citrus germplasm with yin-tonifying medicine that tonifies and nourishes yin fluid, is used to strengthen the body’s resistance to TCM in Southwest China [157].

#### 2.9.2. MOAs of the Extract

PMD extract is rich in flavonoids and phenolics that might ameliorate cognitive dysfunction in β- amyloid-induced rats. Research proved that oral consumption of a flavonoid-rich yuzu extract normalized insulin signaling and improved energy and glucose metabolism in the hippocampus of β-amyloid-infused rats [47].

#### 2.9.3. MOAs of Ingredients

Research indicated that the total phenolic compound content of PMD was 47.8 ± 0.45 mg/100 g, including rutin (4.9 ± 0.7), rutin hydrate (4.2 ± 0.1), narirutin (29.2 ± 3.1), naringin (13.1 ± 1.5), apigen-7-glucoside (17.8 ± 1.5), hesperidin (38.1 ± 2.7), quercetin (1.9 ± 0.2), tangeretin (0.6 ± 0.1), and other ingredients [158,159,160].

Rutin, as a multifunctional biological molecule, can treat and prevent Alzheimer’s disease by regulating various pharmacological mechanisms, such as antioxidant mechanisms, mechanisms related to metal chelation, anti-inflammatory mechanisms, Aβ mechanisms, formation of Aβ, Aβ aggregation and toxicity, and acetyl cholinesterase (AChE) mechanisms [158].

In vitro bioassays have revealed that narirutin is a high-affinity BACE1 inhibitor and extends its applicability as a functional food in AD management [161]. Studies have shown that treatment with naringin (100 mg/kg) enhanced the autophosphorylation of CaMKII and the phosphorylation of the α-amino-3-hydroxy-5-methyl-4-isoxazole propionic (AMPA) receptor in the APPswe/PS1dE9 transgenic mouse model of AD [96].

In addition, hesperidin improved cognitive function and attenuated oxidative stress and apoptosis in an AlCl_3_-induced rat model [97]. It also showed significantly attenuated β-amyloid deposition and TGF-β immunoreactivity, activated Akt/Nrf2 signaling, and inhibited RAGE/NF-κB signaling in the brains of APP/PS1 mice [98,99,162].

Furthermore, Zaplatic E et al. suggested that quercetin has a potential protective effect against neurodegenerative diseases, including AD. On the one hand, the molecular mechanisms of quercetin in attenuating AD are antioxidant activities, mainly involving the nuclear factor (erythroid-derived 2)-like 2 pathway, paraoxonase-2 pathway, JNK-mediated pathways, PKC pathway, MAPK signaling cascades, and PI3K/Akt pathway. On the other hand, the neuroprotective properties of quercetin were able to inhibit Aβ aggregation, NFT formation, amyloid precursor protein (APP) cleaving enzyme (BACE1), and acetylcholinesterase (AChE) [163].

Moreover, limonene (+) from PMD essential oil has a neuroprotective function against the neurotoxicity of Aβ42 [100].

### 2.10. Piper nigrum L.

#### 2.10.1. Traditional Functional Features

*Piper nigrum* L., is widely distributed in tropical regions, and its fruits are used for spices and seasonings as a functional food. *P. nigrum*, an interior-warming medicine in TCM, can dispel cold from the stomach and eliminate phlegm [164].

#### 2.10.2. MOAs of the Extract

Modern pharmacological research has been founded on broad biological activities, including antioxidant, antimicrobial, and insecticidal activities [165,166]. Mahdy K et al. revealed that the total plant extracts of *P. nigrum* reduced oxidative stress and ameliorated neurodegeneration characteristics by significantly increasing Ach and serum total antioxidant capacity (TAC) and SOD and significantly decreasing AchE, MDA, and NO in AD-induced rats [48]. The methanolic extract of P. nigrum (50 and 100 mg/kg) restored the activities of SOD and Catalase (CAT) and increased GPX activity in the hippocampus of Aβ (1–42)-treated rats. Simultaneously, as an antioxidant agent, both doses of the methanolic extract (50 and 100 mg/kg) decreased the protein carbonyl and MDA levels and increased GSH in hippocampal homogenates [49].

#### 2.10.3. MOAs of Ingredients

GC-MS and LC-ESI-MS analyses revealed that β-caryophyllene (51.12%) is the dominant component of P. nigrum essential oil (EO) [167].

Liu et al. demonstrated that β-caryophyllene can alleviate the Alzheimer-like phenotype by inhibiting inflammation and reducing the β-amyloid burden. Their research showed that β-caryophyllene prevented cognitive impairment in APP/PS1 mice by reducing the β-amyloid burden in both the hippocampus and the cerebral cortex. In addition, β-caryophyllene reduced the protein levels of COX-2 and the mRNA levels of the proinflammatory cytokines TNF-α and IL-1β in the cerebral cortex. This study also explored the possibility that in an in vivo experiment, β-caryophyllene ameliorates memory impairment through CB2 receptor activation and the PPARγ pathway in APP/PS1 mice for the treatment of AD [106]. It is well known that α -synuclein forms amyloid aggregates in vitro in the presence of some metal ions or alcohol. A large number of experiments show that the mixture of curcumin and β-caryophyllene can not only inhibit the aggregation of α-synuclein, but also almost completely decompose the preformed aggregates at a relatively low concentration [168].

In addition, piperine, an alkaloid present in black pepper (*Piper nigrum*), has been shown to have anti-inflammatory, antioxidation, cholinergic neuronal transmission, antidepressant, and antipyretic effects [169].

Nazifi M et al. showed that piperine significantly ameliorated cognitive deficits by attenuating oxidative status compared to AD- or AD-Tween-treated groups [108]. Piperine at various doses of 5, 10, and 20 mg/kg significantly improved memory impairment and neurodegeneration in the hippocampus, partly associated with the decrease in lipid peroxidation and the acetylcholinesterase enzyme [109]. In vitro AChE inhibition demonstrated that piperine showed greater AChE inhibition with an IC50 of 76.6 ± 0.08 μg/mL in the SH-SY5Y cell model. Research also found that combined curcumin and piperine inhibited and disaggregated fibril formation and Aβ-induced oxidative damage compared to the Aβ group [107] (Figure 2).

### 2.11. Ganoderma lucidum (Leyss.ex Fr.) Karst.

#### 2.11.1. Traditional Functional Features

*Ganoderma lucidum* (*Leyss.ex Fr.*) *Karst.* is also known as “the mushroom of immortality”, a basidiomycete white rot microfungus used extensively in China for 2000 years. In fact, during ancient times, *G. lucidum* was a qi-tonifying medicine that tonifies health for treating qi deficiency [170].

#### 2.11.2. MOAs of the Extract

It should be mentioned that Ganoderma lucidum displays numerous pharmacological effects, mainly including antibacterial, antiaging, antiviral, antiangiogenic, antifibrotic, antioxidative, anti-inflammatory, and analgesic properties [171,172].

Alleviate nerve injury and neurotoxicity: Ganoderma lucidum can improve behavior disorders associated with a decrease in the levels of the Aβ-40 protein and an increase in the levels of ApoA1, ApoE, and Syt1 [173]. Further studies elucidated that the aqueous extract of *G. lucidum* attenuated the phosphorylation of c-Jun, N-terminal kinase, c-Jun, and p38 MAP kinase, thus blocking Abeta-induced synaptoxicity [174].

Anti-inflammatory effects: *Ganoderma lucidum* ethanol extract (EGL) in the treatment of neurodegenerative diseases inhibited the NF-κB and TLR signaling pathways, thereby reducing the inflammatory response in activated microglia. And an in vitro study showed that EGL exhibited a potential neuroprotective effect by blocking IκB degradation and inhibiting TLR4 and MyD88 expression, which exerted its anti-inflammatory effects in LPS-stimulated BV2 cells [175].

Antioxidative effects: A study by Lee et al. revealed that EGL augmented the cellular antioxidant defense capacity by increasing the expression and phosphorylation of Nrf2 and HO-1 in a C2C12 myoblast cell line [118] (Figure 1).

#### 2.11.3. MOAs of Ingredients

Mechanistic studies clarified that *Ganoderma lucidum* polysaccharide promotes cognitive function and neural progenitor proliferation by activating FGFR1 and inhibiting ERK and AKT cascades [176]. Compared to normal mice, *Ganoderma lucidum* triterpenoids reduced the hippocampus of APP/PS1 mouse brain damage by inhibiting apoptosis (Bax, Bcl2, and caspase 3/cleaved caspase 3), relieving oxidative damage (Nrf2, NQO1, and HO1), and inactivating the increase in ROCK1 and ROCK2 [177]. Lu SY et al. showed that aromatic constituents from Ganoderma lucidum exerted remarkable anti-inflammatory activity with IC50 values ranging from 4.68 to 15.49 μM in RAW264.7 macrophages [176] (Figure 2).

### 2.12. Puerarialobata (Willd.) Ohwi

#### 2.12.1. Traditional Functional Features

*Puerarialobata* (*Willd.*) *Ohwi* is a pungent-cool exterior-releasing medicine in Shennong Bencao Jing common during the Western Han Dynasty (206 BC–8 AD). In the traditional sense, the main effect of Pueraria is to release flesh, clear heat, engender fluid, and outthrust rashes [178].

#### 2.12.2. MOAs of the Extract

Existing pharmacological experiments show that Puerariae Radix offers neuroprotective effects, cardiovascular protection, and osteonecrosis prevention, etc. [116]. To clarify the mechanism of Puerariae Radix for neuroprotective effects, Huang HJ et al. designed C57BL/6J mice that received a direct infusion of soluble oligomeric A to evaluate the extract of Puerariae Radix alleviation of Aβ deposition in the rat hippocampus. Experimentally, it was observed that Puerariae Radix aqueous extract decreased Aβ deposition, tau protein phosphorylation, inflammation, and loss of noradrenergic and serotonergic neurons, while increasing synaptophysin- and insulin-degrading enzymes against the toxicity of oligomeric Aβ [54].

#### 2.12.3. MOAs of Ingredients

Puerarin, isolated from *Puerariae Radix*, has the potential to cure angiocardiopathy and cerebrovascular diseases [179]. The effect of puerarin in the prevention of Aβ-induced neurotoxicity through the inhibition of neuronal apoptosis was also examined. Experiments revealed that treatment of PC12 cells with puerarin dose-dependently ameliorated Bax expression and cytochrome c release and increased P-Akt, Bcl-2, and p-Bad expression [117].

Additionally, in an in vivo experiment, puerarin may improve cognitive performance by activating the Akt/GSK-3β signaling pathway associated with a decrease in the levels of GSK-3β and an increase in the levels of Akt in the hippocampus of APP/PS1 mice [179]. Moreover, in vivo evidence indicates that puerarin can serve as a treatment for age-related neurodegenerative disorders by attenuating tau hyperphosphorylation in D-galactose rats [115].

### 2.13. Ziziphi Spinosae Semen

#### 2.13.1. Traditional Functional Features

*Ziziphus jujuba Mill.* (SJM) mainly grows in the inland areas of northern China, which has a saying that “thorns are everywhere” in ancient China. SJM, a famous heart-nourishing medicine in China, nourishes the yin blood of the heart to calm the mind in TCM [180].

#### 2.13.2. MOAs of the Extract

SJM is a promising MFH item with various pharmacological actions, including anti-inflammatory, anticomplementary activity, hematopoiesis, antioxidant stress, and anticancer properties [181]. In modern pharmacology research experiments, SJM is widely used to alleviate AD-like symptoms. There is evidence that administering SJM orally to 5XFAD mice ameliorated memory impairments by elevating plasmin levels and activity in hippocampal slices from 5XFAD mice [55]. Other studies have demonstrated that SZS ameliorates Aβ-induced LTP deficits through BDNF/TrkB signaling and stimulates plasmin activity in the hippocampus of CD-1 mice [56]. Recent studies suggest that the flavonoid extract of SJM can reduce Aβ-induced toxicity in Caenorhabditis elegans [57].

#### 2.13.3. MOAs of Ingredients

Spinosin is a C-glycoside flavonoid derived from the seeds of SJM that has pharmacological activities, such as anxiolytic and hypnotic effects, and ameliorates memory impairment [182] (Figure 5). Subchronic disease treatment with spinosin (5 mg/kg) activated the ERK-CREB-BDNF signaling pathway to treat cognitive dysfunction in the hippocampus [174]. Another in vitro study using Neuro-2a cells (N2a/WT) and N2a/APP695 cells showed that spinosyn markedly reduced Aβ1–42 production by activating the Nrf2/HO-1 pathway [119]. In addition, spinosin might be beneficial to treat learning and memory deficits through the regulation of oxidative stress, inflammatory processes, apoptotic programs, and plasmin activity [120,121,122].

### 2.14. Gastrodia elata Blume

#### 2.14.1. Traditional Functional Features

*Gastrodia elata Blume*, commonly called Chijian or Ming tian ma in Chinese, is considered a liver-pacifying and wind-extinguishing medicine from the tuber of *Gastrodia elata Blume* (Orchidaceae) [183].

#### 2.14.2. MOAs of the Extract

Studies in vivo demonstrated that *Gastrodia elata* significantly reduced the number of amyloid deposits, increased choline acetyltransferase expression, and decreased the activity of acetylcholinesterase [62]. There is further evidence that *Gastrodia elata* extract (50, 250, and 500 mg/kg) may potentially ameliorate cognitive impairments caused by neuronal cell death in amyloid-β peptide (Aβ)-treated PC12 cells [61].

#### 2.14.3. MOAs of Ingredients

Gastrodin (GAS) is an active constituent extracted from *Gastrodia elata* (Figure 4). Liu Y and other review articles found that GAS has potential therapeutic effects on a variety of central nervous system diseases, including AD, PD, affective disorders, cerebral ischemia/reperfusion, and epilepsy. Its extensive pharmacological activities and principal mechanisms include regulating mitochondrial cascades, antioxidant and anti-inflammatory functions, modulating neurotransmitters, suppressing microglial activation, and upregulating neurotrophins [184].

A molecular analysis showed that GAS could effectively treat BCCAO-induced vascular dementia (BCK) by targeting the formation of Aβ-related proteins (Aβ1–40/42, APP, and β-site APP-cleaving enzyme 1) and inhibiting autophagy (Beclin-1, LC3-II, and p62) and apoptosis (Bax/Bcl-2 and P38 MAPK) in hippocampal neurons [127]. More importantly, GAS promoted Sirt3 upregulation and NOX-2 downregulation, acting on activated microglia in neuroprotection [185].

### 2.15. Lycium barbarum L.

#### 2.15.1. Traditional Functional Features

*Lycium barbarum* L. (a “berry-type” fruit of the plant *Lycium barbarum*), a yin-tonifying medicine belonging to the Solanaceae family, is recorded in Shennong Ben Cao Jing (ca 100 AD) [186].

#### 2.15.2. MOAs of the Extract

In recent years, *Fructus lycii* and its phytochemical components have been increasingly reported as promising anti-AD drugs with key pathological events, such as oxidative stress, anti-immune, antiapoptosis, and antinecrosis effects [187].

Extracts from *Fructus lycii* mitigated neuronal loss and behavioral impairments against the toxicity of fibrillar Abeta (1–42) and Abeta (25–35) fragments. An in vitro study revealed that extracts of *Fructus lycii* (100 μg/mL) ameliorated caspase-3 activity up to 11.8% compared with the Aβ peptide-treated group. In addition, pretreatment with extract isolated from *Fructus lycii* at 100 μg/mL markedly downregulated the protein levels of JNK, c-Jun, and β-actin, while the protein levels of total JNK, total c-Jun, and β-actin were unchanged after treatment with LBA [63] (Figure 1).

#### 2.15.3. MOAs of Ingredients

Gao Y et al. showed that there are many antiaging active ingredients in Fructus lycii, such as Fructus lycii polysaccharides, β-sitosterol, caffeic acid, and zeaxanthin [188]. Studies have shown that the ability of organisms to respond to oxidative stress is intricately related to their aging and longevity [189]. Fructus lycii polysaccharides are a good antioxidant [190]. A systematic review and meta-analysis indicated that dietary zeaxanthin may be protective against age-related macular degeneration [191]. Habtemariam S. et al.’s mini review revealed that the antiAD therapeutic potential of caffeic acid is associated with antioxidant effects, specific anti-inflammatory mechanisms in the brain, and the various processes of β-amyloid formation [192].

### 2.16. Alpinia officinarum Hance/Mosla chinensis Maxim

#### 2.16.1. Traditional Functional Features

*Alpinia officinarum Hance* (AOH), an interior-warming medicine native to China that can warm the stomach and relieve vomiting, dispel cold, and relieve pain is also known as lesser galangal. AOH, which belongs to the Zingiberaceae family, is indigenous to Southeast China (Guangdong, Guangxi, Hainan) [193].

*Mosla chinensis Maxim.* (MCM), a pungent-warm exterior-releasing medicine, can promote diaphoresis and release to the exterior and resolve dampness, thus harmonizing the spleen and stomach and promoting water metabolism [21].

#### 2.16.2. MOAs of Ingredients

AOH is a dietary product with medicinal applications and also possesses a wide range of pharmacological effects, including angiogenesis, antimicrobial, anticancer, anti-inflammatory, vasorelaxation, and antioxidant activities [194]. Diarylheptanoids are considered as principal phytochemical constituents from the rhizome of A. officinarum, including apigenin and galangin [192]. Apigenin, a pharmacologically active agent, is also widely distributed in herbs (*Elsholtzia rugulosa*) [195].

Previous studies have proven that apigenin resists Aβ-induced toxicity, has anti-inflammatory and antioxidant effects, alleviates learning and memory deficits, and depresses neuronal apoptosis [196,197].

Inhibiting the apoptosis of neurons: Studies have shown that apigenin protects neuronal cells from injury by suppressing the phosphorylation of inducible nitric oxide synthase, cyclooxygenase-2 protein, p38 mitogen-activated protein kinase (MAPK), and c-Jun N-terminal kinase (JNK) in primary microglial cells [103]. Another study showed that apigenin could exert neuroprotection against Aβ-induced toxicity in the presence of copper mainly through inhibiting the p38 MAPK signaling pathway and SAPK/JNK pathway and depressing neuronal apoptosis in vitro [101].

Neuro-immunomodulatory: Dourado NS et al. evaluated the neuroimmunomodulatory and neuroprotective effects of apigenin on glial cells and neurons of Wistar rats. Treatment with apigenin preserves neuron and astrocyte morphology by reducing the expression of OX42, IL-6, and gp130. In addition, apigenin alone and after an inflammatory stimulus with IL-1β also increased the expression of BDNF to exert anti-inflammatory and neuroprotective effects [104]. In addition, apigenin also downregulated BACE1 and β-CTF levels, affecting APP processing and preventing the Aβ burden. Moreover, apigenin restored the neurotrophic ERK/CREB/BDNF pathway in the cerebral cortex for the prevention and/or therapy of AD [102].

Improving cholinergic neuronal transmission: Acetylcholinesterase (AChE) inhibitors are an important class of drugs for the treatment of AD, such as rivastigmine, galantamine, and huperzine. Galangin is a major flavonoid found in Rhizoma Alpiniae Officinarum with the strongest inhibitory effect on AChE activity (56.53 ± 0.03) [105] (Figure 5).

### 2.17. Curcuma longa L.

#### 2.17.1. Traditional Functional Features

*Curcuma longa* L., a blood-activating analgesic medicine extensively cultivated in India and China, has been widely used as a spice in foods. It can eliminate blood stasis, promote the flow of qi, stimulate menstrual discharge, and relieve pain [198].

#### 2.17.2. MOAs of Ingredients

*Curcuma longa* L. inhibits anti-inflammatory, antihuman immunodeficiency virus, antibacterial, and antioxidant effects [199,200].

Curcumin (Cur), a potent antiamyloid natural polyphenol, is derived from the root of RCL (Figure 3). Studies have shown that Cur can effectively modify AD pathology by preventing the formation and accumulation of Aβ, tau inhibition, copper-binding and cholesterol-lowering abilities, anti-inflammatory activity and modulation of microglia, acetylcholinesterase inhibition, regulation of the insulin signaling pathway, and antioxidant activity [201,202]. For example, Cur directly bound to PPARγ increased the transcriptional activity and protein levels of PPARγ and inhibited the nuclear factor kappa B (NF-κB) signaling pathway, indicating that the beneficial effects of Cur on AD are attributable to the suppression of neuroinflammation in APP/PS1 mice [110].

Da Costa et al., in a systematic review, indicated that Cur supplementation, a promising approach in AD, reversed neurotoxic and behavioral damage in both in vivo (Sprague–Dawley rats, APPswe/PS1dE9 and 5xFAD transgenic mice) and in vitro (PC12 cells, SK-N-SH cells, and neonate rat cells) models of AD [203]. For instance, Cur (50 mg/kg) also provided antiamyloid and neuroprotective outcomes in 5xFAD mice [111].

Aromatic-turmerone is an analog of curcumin and is rich in *Curcuma longa*. Accumulating evidence has demonstrated that aromatic-turmerone impairs the Aβ-induced inflammatory response of microglial cells by inhibiting the NF-κB, JNK, and p38 MAPK signaling pathways in hippocampal HT-22 cells [112]. Moreover, aromatic-turmerone significantly limited brain damage by inhibiting TLR4 and lowered the release of inflammatory mediators [113]. Further evidence was provided that aromatic- turmerone prevented cleaved caspase-3, while neither the level of ROS nor the mitochondrial membrane potential was affected in cerebellar granule neurons [114].

## 3. Other MFH with Potential AntiAD Activity

In addition to the abovementioned more comprehensive and systematic preclinical and clinical studies on MFH, there are other studies that show positive effects on AD. Among them, some MFH substancess, such as *Alpinia officinarum Hance* [103], *Mosla chinensis Maxim.* [101,104], *Curcuma longa* L. [110,111], *Glycyrrhiza uralensis Fisch.* [123,124,125], *Raphanus sativus* L. [126], and *Cornus officinalis Siebold & Zucc.* [128], focus only on the pharmacological effects of their active ingredients against AD, as shown in Table 4. However, other MFH items have only conducted in vivo/in vitro pharmacological experimental studies on their crude extracts against AD, including *Morus alba* L. [50,51,52,53], *Cinnamomum cassia* (L.) *J. Presl* [58,59,60], *Houttuynia cordata thunb.* [64], *Cassia obtusifolia* L. [65,66],, *Ziziphus jujuba Mill.* [67,68,69], *Dendrobium nobile Lindl.* [70,71,72], and *Piper longum* L. [73,74] (Figure 1 and Figure 2). It is worth developing a better experimental design and conducting clinical research in the future.

## 4. Discussion and Conclusions

Historically, Chinese ancestors developed many kinds of TCM through long-term practice and summarized their application in treating disease [204]. They also realized that many of these TCMs could be eaten as functional foods (TCM nutrition) in their daily lives, which reflects the theory of MFH [8,13,205].

Since the 20th century, chronic diseases have become the main cause of global morbidity and mortality. A very important reason for this is the imbalance of nutrition and suboptimal lifestyle behaviors, leading to such diseases as AD [206,207]. In recent years, limited by the complexity of AD pathology, new drugs targeting amyloid-β peptide (Aβ) or tau proteins have failed to show significant clinical benefits, such as ClinicalTrials.gov identifier: CAD106 (NCT01097096), ACC-001 (NCT01284387), and AFFITOPE AD02 (NCT01117818 CAD106) [208,209]. Consequently, the development of new drugs and new treatment methods have encountered considerable obstacles and bottlenecks. Interestingly, some TCMs are used as functional foods with brain targets, nutritional benefits, and long-term applications, adjusting diet nutrition to prevent the occurrence of AD, such as *Salvia miltiorrhiza Bunge*, *Cinnamomum cassia* (L.) *J.Presl*, *Zingiber officinale Roscoe*, and *Panax ginseng C.A. Mey.* [210,211] (Figure 1 and Figure 2). Taking MFH substances as the primary functional foods can ensure the safety and therapeutic effect of functional food. At the same time, good social and economic benefits can also be obtained by using the cost and technical advantages of TCM, coupled with the progress of the modern food industry [212]. Therefore, MFH foods are used as a dietary intervention to provide higher specialist guidance for AD.

However, although pharmacological research and clinical research have been devoted to proving the safety and effectiveness of MFH items in treating AD, the MOAs of MFH and their active components against AD are still unclear [211,213]. Fortunately, TCM transcriptomics, metabonomics, and protein omics, as well as TCM systematic pharmacology, are in full swing. Future research can combine multiomics and systems pharmacology to explore the pharmacokinetics of the multiple ingredients in these MFH substances [214,215]. In addition, most studies on the mechanism and efficacy of MFH are still at the animal and cellular levels; therefore, clinical research is scarce. Future clinical research should be performed to discover novel antiAD agents of MFH foods. Moreover, given the diversity of AD pathology, MFH antiAD research should also focus on its multicomponent and multitarget mechanisms in a holistic way.

MFH originated in China, but is now favored by people worldwide in the daily diet to prevent the occurrence and development of AD. The rapid development of modern science and technology should provide an advanced theory and a wider field of vision for explaining the advantages of traditional MFH theory in AD. Particularly in recent years, overcoming chronic diseases requires more basic and clinical research to develop new drugs. As a renewable and promising resource, MFH should be a resource to take ongoing effective measures to develop and produce new antiAD drugs or health care products.

## Figures and Tables

**Figure 1 molecules-27-00901-f001:**
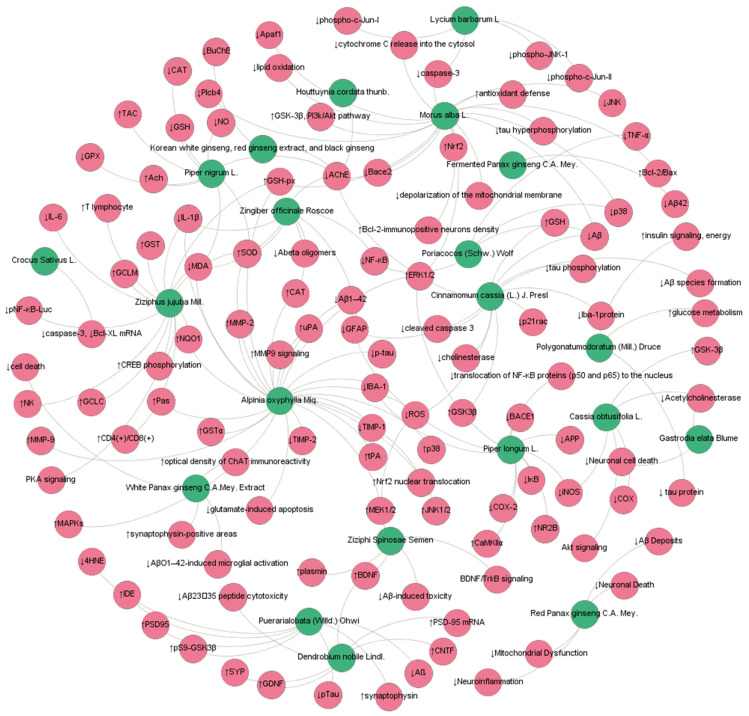
MFH antiAD network mechanism diagram. (Blue nodes represent MFH, and red nodes represent targets/pathways).

**Figure 2 molecules-27-00901-f002:**
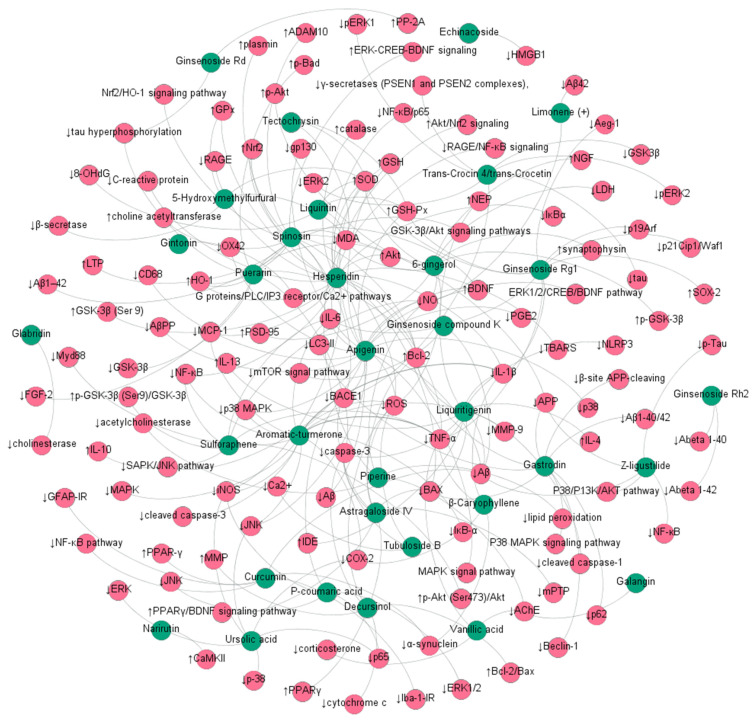
The ingredients of MFH antiAD network mechanism diagram. (Blue nodes represent ingredient, and red nodes represent targets/pathways).

**Figure 3 molecules-27-00901-f003:**
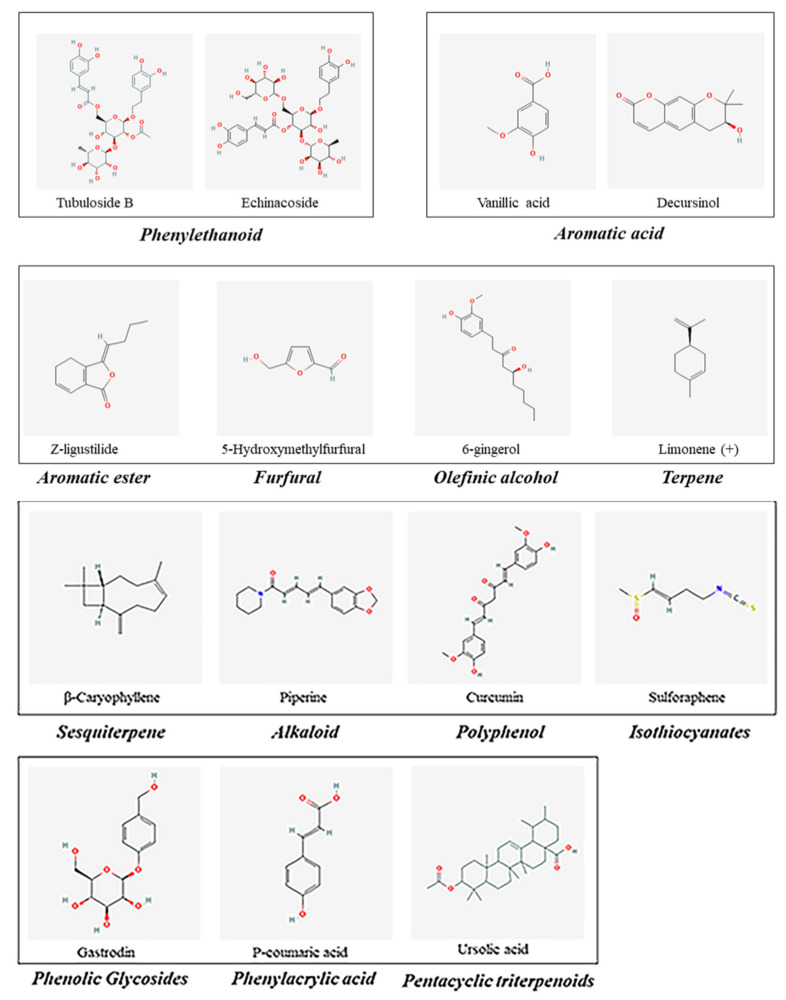
Other active ingredients in MFH.

**Figure 4 molecules-27-00901-f004:**
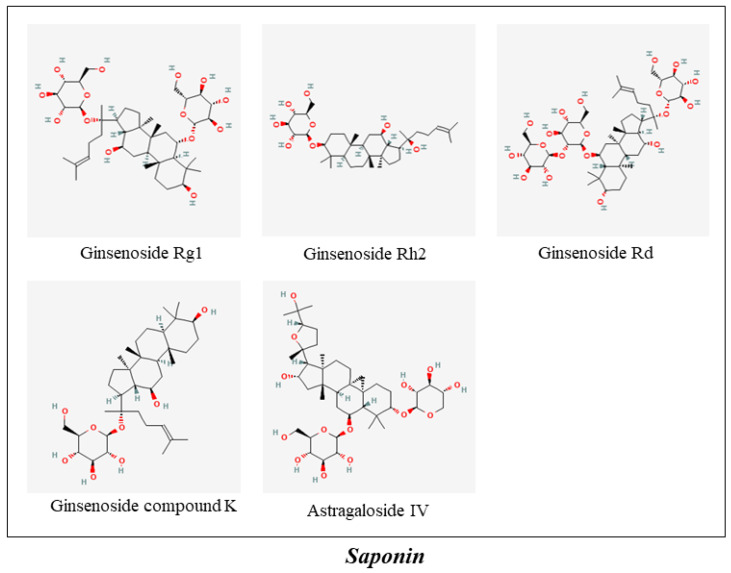
The ingredients of Saponin in MFH.

**Figure 5 molecules-27-00901-f005:**
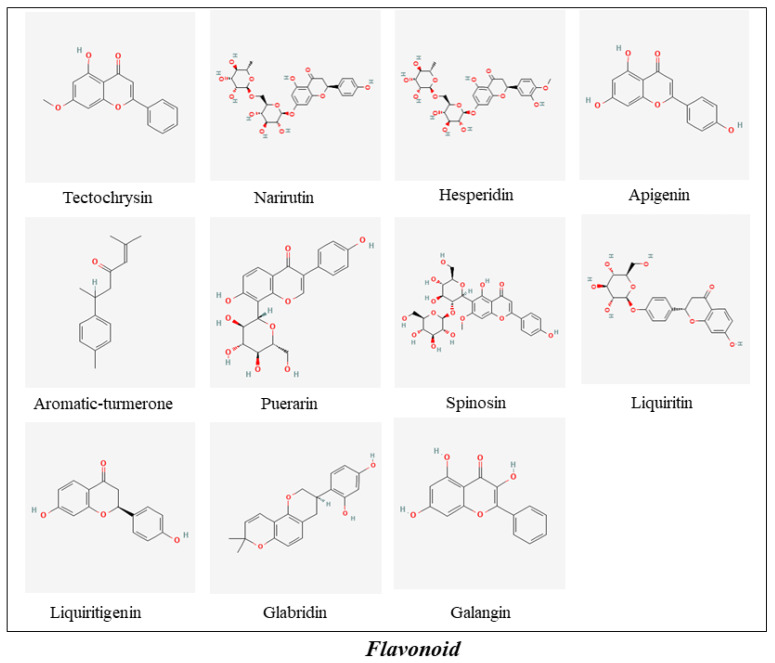
The ingredients of Flavonoid in MFH.

**Table 1 molecules-27-00901-t001:** The clinical evidence of prevention and treatment of Alzheimer’s disease using MFH.

No.	Registration No	Number of Subjects	Trial Period	Intervention Model	Intervention Group	Controlled Group	Findings/Mechanism	References
1	NCT00391833	77 patients	12 weeks	An open-label study	*Panax ginseng C.A. Mey.*		Panax ginseng is clinically effective in the cognitive performance of AD patients.	[14]
2		61 patients	12-week	An open-label trial	*Korean red Panax ginseng C.A. Mey.*	Either donepezil, galantamine, memantine or rivastigmine	Korean red ginseng showed good efficacy for the treatment of Alzheimer’s disease	[15]
3		91 patients	24-week	A 24-week randomized open-label study			The effect of KRG on cognitive functions was sustained for 2-year follow-up, indicating feasible efficacies of long-term follow-up for Alzheimer’s disease.	[16]
4	IRCT138711051556N1	91 patients	22 weeks	A multicenter, randomized, double-blind controlled trial	*Crocus sativus* L.	Donepezil	This phase II study provides preliminary evidence of a possible therapeutic effect of saffron extract in the treatment of patients with mild-to-moderate Alzheimer’s disease.	[17]
5		56 patients	16 weeks	A randomized and placebo-controlled trial		Placebo	Saffron is both safe and effective in mild to moderate AD.	[18]
6		17 patients on crocus and 18 on a waiting list	12 months	1-year single-blind randomized, with parallel groups, clinical trial			Crocus is a good choice for management of aMCImd	[19]
7		24 participants			*Cistanche afghanica Gilli*	Donepezil	Cistanches Herba could improve cognitive and independent living ability of moderate AD patients, slow down volume changes of hippocampus, and reduce the levels of T-tau, TNF-α, and IL-1β.	[20]

**Table 2 molecules-27-00901-t002:** MFH varieties and their functional features [21].

Names		Parts to Be Consumed	Therapeutic Class	Features
Chinese Name	English Name			
Ren shen	*Panax ginseng C.A. Mey.*	Root	Qi-reinforcing medicinal	To reinforce the vital energy, to remedy collapse and restore the normal pulse, to benefit the spleen and lung, to promote the production of body fluid, and to calm the nerves.
Xi hong hua	*Crocus sativus* L.	Stigma	Blood-activating stasis removing medicinal	To activate blood circulation and eliminate blood stasis, to remove heat from blood and counteract toxicity, and to calm the nerves.
Rou cong rong	*Cistanche afghanica Gilli*	Fleshy stem	Yang-reinforcing medicinal	To reinforce the kidney, replenish vital essence and blood, and induce taxation.
Dang gui	*Angelica acutiloba* (*Siebold & Zucc.*) *Kitag.*	Root	Blood-tonifying medicinal	To enrich blood, activate blood circulation, regulate menstruation, relieve pain, and relax bowels.
Huang qi	*Astragalus aaronii* (*Eig*) *Zohary*	Root	Qi-reinforcing medicinal	To reinforce qi and invigorate the function of the spleen.
Fu ling	*Poriacocos* (*Schw.*) *Wolf*	Sclerotium	Diuretic dampness excreting medicinal	To cause diuresis, to invigorate the spleen function, and to calm the mind.
Yi zhi ren	*Alpinia oxyphylla Miq.*	Fruit	Astringent medicinal	To warm spleen, check diarrhea, warm kidney, reduce urine, and secure essence.
Sheng jiang	*Zingiber officinale Roscoe*	Fresh rhizome	Pungent-warm exterior-releasing medicinal	To induce perspiration and dispel cold, to warm the stomach and arrest vomiting, and to resolve phlegm and relieve cough.
Yu zhu	*Polygonatumodoratum* (*Mill.*) *Druce*	Rhizome	Yin-tonifying Medicinal	To nourish yin, promote the production of body fluid, and relieve dryness syndromes.
Gao liang jiang	*Alpinia officinarum Hance*	Rhizome	Interior-warming medicinal	To warm the stomach and relieve vomiting, dispel cold, and relieve pain.
Xiang ru	*Mosla chinensis Maxim.*	Aerial parts	Pungent-warm exterior-releasing medicinal	To promote diaphoresis and release to the exterior, to resolve dampness and harmonize the spleen and stomach, to promote water metabolism and release edema
Hei hu jiao	*Piper nigrum* L.	Fruit	Interior-warming medicinal	To dispel cold from the stomach and to eliminate phlegm.
Ling zhi	*Ganoderma lucidum* (*Leyss.ex Fr.*) *Karst.*	Dried sporocarp	Qi-reinforcing medicinal	To invigorate qi and calm the nerves and relieve cough and asthma.
Sang shen	*Morus alba* L.	Ear of fruit	Yin-tonifying medicinal	To promote the production of body fluid.
Jiang huang	*Curcuma longa* L.	Tuberoid	Blood-activating and stasis-dispelling medicinal	To eliminate blood stasis, promote the flow of qi, stimulate the release of menstruation, and relieve pain.
Ge gen	*Puerarialobata* (*Willd.*) *Ohwi*	Tuberoid	Pungent-warm exterior-releasing medicinal	To relieve fever, to promote the production of body fluid, to facilitate eruption, and to arrest diarrhea.
Suan zao ren	*Ziziphi Spinosae Semen*	Seed	Heart-nourishing tranquillizing medicinal	To replenish the liver, to cause tranquilizations, to arrest excessive perspiration, and to promote the production of body fluid.
Gan cao	*Glycyrrhiza uralensis Fisch.*	Root, rhizome	Qi-reinforcing drugs	To reinforce the function of the spleen and replenish qi, to remove heat and counteract toxicity, to dispel phlegm and relieve cough, to alleviate spasmodic pain, and to moderate drug actions.
Lai fu zi	*Raphanus sativus* L.	Seed	Digestants	To promote digestion and relieve abdominal distension and to relieve cough and resolve phlegm.
Rou gui	*Cinnamomum cassia* (L.) *J.Presl*	Bark	Interior-warming medicinal	To supplement body fire, to reinforce yang, and to lead the fire back to the kidney, to dispel cold and relieve pain, and to activate blood circulation and stimulate menstrual discharge.
Tian ma	*Gastrodia elata Blume*	Stem and tuber	Liver-pacifying and wind-extinguishing medicinal	To extinguish wind and check tetany, to calm liver and subdue yang, to dispel wind, and to free network vessels.
Gou qi zi	*Lycium barbarum* L	Fruit	Yin-tonifying medicinal	To benefit the liver and the kidney, to replenish vital essence, and to improve eyesight.
Yu xing cao	*Houttuynia cordata thunb.*	Aerial parts	Heat-clearing and detoxicating medicinal	To remove toxic heat, to promote the drainage of pus, and to relieve dysuria.
Jue ming zi	*Cassia obtusifolia* L.	Ripe seed	Fire-purging medicinal	To remove heat from the liver, to improve eyesight, and to relax bowels.
Shan zhu yu	*Cornus officinalis Siebold & Zucc.*	Fruit	Astringent medicinal	To replenish the liver and kidney, to restrain seminal discharge, and to relieve collapse.
Da zao	*Ziziphus jujuba Mill.*	Ripe fruit	Qi-reinforcing medicinal	To tonify the spleen and replenish qi, to nourish blood, and to ease the mind.
Shi hu	*Dendrobium nobile Lindl.*	Fresh or dried stems	Yin-tonifying medicinal	Treatment of thirst due to impairment to yin or deficiency of body fluid, loss of appetite with nausea, fever in deficiency conditions after a severe disease, and impaired vision.
Bi bo	*Piper longum* L.	Ear of fruit	Interior-warming medicinal	To warm the interior and expel internal cold

**Table 3 molecules-27-00901-t003:** The mechanism of “Jun medicine” of MFH in TCM formulae against Alzheimer’s disease.

No.	“Jun Medicine” of MFH	TCH Formulae	Dose Range		Controls		Experimental Models		Mechanism	References
			In Vitro	In Vivo	Positive	Negative	In Vitro	In Vivo		
1	*Panax ginseng C.A. Mey.*	Shenma Yizhi Decoction (SMYZD)		2.97, 11.88 g/kg				Wistar rats	↑NG2, ↑Ang1, ↑PDGFR-β	[22]
		Yizhi Qingxin Formula (YZQXF)		0.3, 0.6 mg/kg	Donepezil hydrochloride	Distilled water		Wistar rats	↑IL-10, ↓amyloid-β peptide, ↓TNFα, ↓IL2, ↓IL-6, ↑BDNF, ↑TrkB, ↑BDNF/TrkB pathway, ↑Erk and Akt signaling	[23]
				2.6, 5.2, 10.4 g·kg^−1^		Distilled water		APP/PS1 mice	↑LC3II/LC3I, ↑Beclin1, ↓mTOR, ↑4EBP1, ↓p62, ↑cathepsin D, ↑V-ATPase, ↓Aβ	[24]
		Jangwonhwan	50, 75, 100, 150 µg/mL	400 mg/kg			SH-SY5Y cells	APPswe/PS1De9 mice	↓Abeta (1-42), ↓Abeta (1-40)	[25]
2	*Angelica acutiloba* (*Siebold & Zucc.*) *Kitag.*	Danggui Buxue Tang (DGBXT)	25, 50, 75, 100 µM				Cortical neurons		↑Bcl2/Bax, ↓cleaved-caspase 3/9, ↓PARP	[26]
		Danggui-Shaoyao-San (DGSYS)		1.6, 3.2, 4.8 g/kg				SAMR1 and SAMP8 mice	↑E2, ↑NO, ↑glycine	[27]
				1.6, 3.2, 6.4 g/kg	Aricept	Saline		APP/PS1 mice	↓PEG2, ↓TXB2, ↓LTB4, ↓cPLA2, ↓COX-1, ↓COX-2	[28]
		Fo Shou San (FSS)		1.6, 3.2, 6.4 g/kg	Donepezil	Ultrapure water		APP/PS1 mice	↓LPS, ↑AP, ↓MDA	[29]
3	*Astragalus aaronii* (*Eig*) *Zohary*	Tiaoxin Recipe		0.057 g/d				APPswe/PS1De9 mice	↓amyloid plaque, ↓Aβ1-42	[30]
		Fuzheng Quxie Decoction (FZQXD)		1.3, 2.6 g/kg				SAMP8 mice	↓HIF1α	[31]
4	*Poriacocos* (*Schw.*) *Wolf*	Lingui Zhugan Decoction (LGZGD)		4.8, 2.4 or 1.2 g/kg	Donepezil	Saline		Sprague–Dawley rats	↓TNF-α, ↓IL-1β, ↓Aβ1-42, ↓p-Erk1/2, ↓p-p38, ↓p-NF-κB, ↓p-IκBα, ↓MAPK signaling, ↓NF-κB signaling	[32]

**Table 4 molecules-27-00901-t004:** Extract from MFH on the role and mechanism of Alzheimer’s disease.

No.	Extract of MFH	Dose Range		Controls		Experimental Models		Mechanism	References
		In Vitro	In Vivo	Positive	Negative	In Vitro	In Vivo		
1	*Red Panax ginseng C.A. Mey.*	1, 10, 100, 500, and 1000 μg/mL	100 mg/kg			HT22 cells	5XFAD mice	↓Mitochondrial Dysfunction; ↓Aβ Deposits; ↓Neuroinflammation; ↓Neuronal Death	[33]
	*Fermented Panax ginseng C.A. Mey.*	200 or 400 μg/mL	400 and 800 mg/kg		Saline	HeLa cells	ICR mice; APP/PS1 mice	↓Aβ42	[34]
	*White Panax ginseng C.A. Mey.* extract		100 and 500 mg/kg		Saline		ICR mice	↓cell death; ↓AβO1–42-induced microglial activation; ↑synaptophysin-positive areas; ↑optical density of ChAT immunoreactivity	[35]
	Korean white ginseng, red ginseng extract, and black ginseng	0.5, 2, 4, 6, 8 mg/mL	2 mg/kg		0.9% saline	Spectrophotometric method	ICR mice	↓AChE; ↓BuChE	[16]
2	*Crocus Sativus* L.	1, 10 μM				PC-12 cells		↓caspase-3, ↓Bcl-XL mRNA	[36]
3	*Poriacocos* (*Schw.*) *Wolf*		1.2 g kg^−1^/day				APP/PS1 mice	↓Aβ	[37]
			100, 300, 900 mg/kg	Fluoxetine	0.5% carboxymethyl cellulose		Sprague–Dawley rats	↓p38, ↓NF-κB, ↓TNF-α	[38]
4	*Alpinia oxyphylla Miq.*	80, 120, 160, 200, 240 μg/mL				Cerebral cortices of BALB/c mouse		↓glutamate-induced apoptosis	[39]
		20, 40, 60, 80, 100, 150, 200 μg/mL	30, 60, 100, 150, 200 mg/mL			RSC96 Schwann cells	Sprague–Dawley rats	↑uPA, ↑tPA, ↑MMP-9, ↑MAPKs, ↑Pas, ↑MMP9 signaling, ↑MEK1/2, ↑ERK1/2, ↑JNK1/2, ↑p38, ↑MMP-2, ↓TIMP-1, ↓TIMP-2	[40]
			360 mg/kg	Donepezil	Saline		ICR mice	↓IBA-1, ↓IL-1β, ↓Aβ1–42, ↓p-tau	[41]
			180, 360 mg/kg	Donepezil	Saline		ICR mice	↑GSH-px, ↓MDA, ↓Aβ1–42	[42]
5	*Zingiber officinale Roscoe*		1, 2, 4 g/kg		Saline		Sprague–Dawley rats	↓MDA, ↑SOD, ↑CAT, ↓NF-Κb, ↓IL-1β	[43]
			50, 100, 200 mg/kg		Carboxymethyl cellulose		Wistar rats	↓IL-1β, ↓GFAP	[44]
		0.13, 0.19, 0.25, 0.31 mg/mL				Rat brain		↓AChE, ↓MDA	[45]
		0.02, 0.1, 0.2 mg in 2 Μl of DMSO				Rat hippocampal		↓Abeta oligomers	[46]
6	*Polygonatumodoratum* (*Mill.*) *Druce*		3% lyophilized 70% ethanol extracts				Hippocampal Slices of Sprague–Dawley rats	Insulin signaling, energy↑, glucose metabolism↑, tau protein↓	[47]
7	*Piper nigrum* L.		187.5 or 93.75 mg/kg	Aluminum chloride			Rats	↑Ach, ↑TAC, ↑SOD, ↓AchE, ↓MDA, ↓NO	[48]
			50 mg/kg, 100 mg/kg	Diazepam			Wistar rats	↓SOD, ↓GPX, ↓CAT, ↓GSH, ↓MDA	[49]
8	*Morus alba* L.	0.1, 1, 10 μg/mL	20 mg/kg, 100 mg/kg, and 500 mg/kg	Donepezil	Saline	Hippocampus of Sprague–Dawley rats	Sprague–Dawley rats	↑GSK-3β, PI3k/Akt pathway, ↓tau hyperphosphorylation, ↑Bcl-2/Bax, ↓depolarization of the mitochondrial membrane, ↓cytochrome C release into the cytosol, ↓caspase-3	[50]
			2, 10, 50 mg/kg	Donepezil			Wistar rats	↓AChE, ↑Bcl-2-immunopositive neurons density, ↑GSH-Px	[51]
			50, 100, 500 mg/kg				SAMR1 and SAMP8 mice	↓lipid oxidation, ↑antioxidant defense, ↓p38, ↓JNK, ↑ERK1/2, ↑Nrf2, ↓Aβ	[52]
		200 μg/mL				PC12 cells		↓Apaf1, ↓Bace2, ↓Plcb4	[53]
9	*Puerarialobata* (*Willd.*) *Ohwi*		340 mg/kg		Saline		C57BL/6J mice	↓4HNE, ↑pS9-GSK3β, ↓Aß, ↑IDE, ↓pTau, ↑PSD95, ↑synaptophysin	[54]
10	*Ziziphi Spinosae Semen*	3, 30 μg/mL				Hippocampal of 5XFAD mice		↑plasmin	[55]
		10, 30, 100 μg/mL				Hippocampal Slices of CD-1 mice		↑BDNF, BDNF/TrkB signaling, ↑plasmin	[56]
		20, 50, 100, 200, 400 µg/mL				PC12 cells	Caenorhabditis elegans	↓Aβ-induced toxicity, ↓ROS	[57]
11	*Cinnamomum cassia* (L.) *J. Presl*	0.75 mg/mL	100 µg/mL		Water		5XFAD mice	↓Aβ species formation	[58]
		500 μm	100 mg/kg			BV-2 cells	5XFAD mice	↓ROS, ↓p21^rac^, ↑GSH, ↓GFAP, ↓Iba-1protein, ↓cleaved caspase 3, ↓tau phosphorylation, ↓Aβ	[59]
			50 mg/kg		Saline		Sprague–Dawley rat	↓AChE, ↑GSK3β, ↓cholinesterase	[60]
12	*Gastrodia elata Blume*	0.01, 0.05, 0.5, 1, 5, 10, 20 and 30 mg/mL	50, 250, 500 mg/kg		Scopolamine	PC12 cells	Sprague–Dawley rats	↓Neuronal cell death	[61]
			500, 1000 mg/kg		0.5% cellulose		Rats	↓Acetylcholinesterase	[62]
13	*Lycium barbarum* L	10^−4^, 10^−3^, 10^−2^, 10^−1^, 1, 10, 10^2^ μg/mL				Neurons of Sprague–Dawley rats		↓caspase-3, ↓phospho-JNK-1, ↓phospho-c-Jun-I, ↓phospho-c-Jun-II	[63]
14	*Houttuynia cordata thunb.*		400 mg/kg	Donepezil		Cortical neurons	Male ICR mice	↓AChE	[64]
15	*Cassia obtusifolia* L.		12.5, 25, 50, 100 mg/kg				ICR mice	↓acetylcholinesterase	[65]
		1, 10 μg/mL	50 mg/kg			Hippocampal	CD-1 mice	↑GSK-3β, Akt signaling, ↓iNOS, ↓COX	[66]
16	*Ziziphus jujuba Mill.*	0.75, 1.5, 3 mg/mL				Astrocytes from postnatal SD rat		↑NQO1, ↑GCLC, ↑GCLM, ↑GST	[67]
		0.75, 1.5, and 3.0 mg/mL				PC12 cells		PKA signaling, ↑CREB phosphorylation	[68]
		0.375, 0.75, 1.5, and 3.0 mg/mL				RAW 264.7 macrophages		↓IL-1β, ↓IL-6, ↓pNF-κB-Luc	[69]
17	*Dendrobium nobile Lindl.*		40 mg/kg		Distilled water		Kunming mice	↑BDNF, ↑GDNF, ↑CNTF	[70]
		0.025, 0.25, 2.5 mg/L				Primary neuronal cultures		↑PSD-95 mRNA, ↑SYP	[71]
		3.5, 35, 350 ng/mL				Hippocampus tissues		↓Aβ23-35 peptide cytotoxicity,	[72]
18	*Piper longum* L.		10 μL/g		0.5% carboxymethyl cellulose		C57BL/6J mice	↑NR2B, ↑ERK1/2, ↑CaMKIIα	[73]
		0.5, 1.0, 2.5 μM	1.5, 3 mg/kg		DMSO + PBS	BV-2 cells	ICR mice	↓APP, ↓BACE1, ↓COX-2, ↓iNOS, ↓translocation of NF-κB proteins (p50 and p65) to the nucleus, ↓IκB, ↓GFAP, ↓ IBA-1	[74]

**Table 5 molecules-27-00901-t005:** Active ingredients from MFH on the role and mechanism of Alzheimer’s disease.

No.	Ingredients of MFH	Category	Source	Dose Range		Controls		Experimental Models		Mechanism	References
				In Vitro	In Vivo	Positive	Negative	In Vitro	In Vivo		
1	GinsenosideRg1	Saponin	*Panax ginseng C.A. Mey.*		20 mg/kg·d		Saline		SD rats	↑SOX-2, ↓Aeg-1, ↑GSH-Px, ↑SOD, ↓IL-1β, ↓IL-6, ↓TNF-α, p53, ↓p21Cip1/Waf1, ↓p19Arf	[75]
2	Ginsenoside Rh2	Saponin		1.5, 2.5, 3.5 μM	10 mg/kg		Saline	Cortex neurons	tg2576 mice	↓Abeta 1-40, ↓Abeta 1-42	[76]
3	Ginsenoside Rd	Saponin		2.5 or 5 μmol/L	10mg/kg		Saline	Cortical neurons	Sprague–Dawley rats	↓tau hyperphosphorylation, ↑PP-2A	[77]
4	Gintonin	Lysophosphatidic Acids-Protein Complexes		0.1, 0.3, 1.0, 3.0, 10 μg/mL				B103 cells		transient Ca^2+^ mobilization, Gproteins/PLC/IP3 receptor/Ca^2+^ pathways	[78]
5	Ginsenoside compound K	Saponin		50, 20, 10, 1 and 0 μM				Primary astrocytes		↓Aβ, ↓mTOR signal pathway	[79]
6	Trans-Crocin 4/trans-Crocetin		*Crocus sativus* L.	0.1, 1, 10, 100, 1000 μM				SH-SY5Y cells; PC12 cells		↓BACE1, ↓γ-secretases (PSEN1 and PSEN2 complexes), ↓tau, ↓GSK3β, ↓ERK2, ↓pERK1, ↓pERK2	[80]
7	Tubuloside B	Phenylethanoid	*Cistanche afghanica Gilli*	1, 10, or 100 mg/L				SH-SY5Ycells		↓ROS, ↓caspase-3, ↓Ca^2+^	[81]
8	Echinacoside	Phenylethanoid			60 mg/kg				C57BL/6 mice	↓HMGB1	[82]
9	Vanillic acid	Aromatic acid	*Angelica acutiloba* (*Siebold & Zucc.*) *Kitag.*		25, 50, and 100 mg/kg				Swiss albinomice	↓AChE, ↓corticosterone, ↓TNF-α	[83]
10	Decursinol				2, 4, and 8 mg				ICR mice	↓Aβ (1-40) induced memory impairment	[84]
				0.01, 0.1, 1.0, 10 μM				PC12 cells		↓ROS, ↑Bcl-2/Bax, ↑MMP, ↓cytochrome c, ↓Caspase-3	[85]
11	Z-ligustilide	Aromatic ester			40 mg/kg		Saline		SPF Wistar rats	↓Aβ, ↓APP, ↓p-Tau, ↓NF-κB	[86]
				0.01, 1, 3, 10, 30 μM					SH-SY5Y cells; PC12 cells	p38/ PI3-K/Akt pathways	[87]
12	Astragaloside IV	Saponin	*Astragalus aaronii* (*Eig*) *Zohary*	25, 50, 100 μM				HT22 cells		↑PPARγ/BDNF signaling pathway	[88]
					10.0 mg/kg		Saline		APP/PS1 mice	↑PPARγ, ↓BACE1	[89]
				10, 25, 50 µM				SK-N-SH cells		↓Bax, ↓caspase-3, ↑Bcl-2, ↓mPTP, ↓ROS	[90]
13	5-Hydroxymethylfurfural	Furfural	*Alpinia oxyphylla Miq.*		15, 150 μg/kg				Kunming mice	↓β-secretase, ↓MDA, ↑GPx, ↑SOD	[91]
14	Tectochrysin	Flavonoid			14 µg/kg	Donepezil			Kunming mice	↑SOD, ↑GSH-px, ↓MDA	[92]
15	6-gingerol	Olefinic alcohol	*Zingiber officinale Roscoe*	40, 80, 120, 200, 300 μM				PC12 Cells		↓ROS, ↓NO, ↓LDH, ↑SOD, ↓MDA, ↑p-Akt, ↑p-GSK-3β	[93]
					6 mg/kg		DMSO		C57BL/6N mice	↑emotional memory deficit	[94]
					10 mg/kg	Donepezil	Saline		ICR mice	↑NGF, ↑PSD-95, ↑synaptophysin	[95]
16	Narirutin	Flavonoid	*Polygonatumodoratum* (*Mill.*) *Druce*		50, 100 mg/kg				APPswe/PS1dE9 mice	↑CaMKII	[96]
17	Hesperidin	Flavonoid			100 mg/kg		Saline		Wistar rats	↓TBARS, ↑GSH, ↑SOD, ↑catalase, ↑GPx, ↓Bax, ↑Bcl-2	[97]
				1, 3 and 9 μg/mL	100 mg/kg		1% CMC	RAW 264.7 cells	APP/PS1 mice	↓Aβ plaques, ↓iNOS, ↓TNF-α, ↓IL-1β	[98]
					20, 40, 80 mg/kg				APP/PS1 mice	↓TNF-α, ↓C-reactive protein ↓MCP-1, ↓NF-κB, ↑Akt, ↑GSK-3β (Ser 9), ↑Nrf2, ↑HO-1, ↓RAGE, ↓IκBα, ↓NF-κB/p65, ↑Akt/Nrf2 signaling, ↓RAGE/NF-κB signaling	[99]
18	Limonene (+)	Terpene		50 μg/mL					Fly Strains	↓Aβ42, ↓NO	[100]
19	Apigenin	Flavonoid	*Mosla chinensis Maxim.*/*Alpinia officinarum Hance*		40 mg/kg		5% CMC-Na		APP/PS1 Mice	↓Aβ, ↓BACE1, ERK1/2/CREB/BDNF pathway	[101]
				0.1, 1.0, 10.0 μM				SH-SY5Y cells		↓AβPP, ↓ROS, ↑GSH, ↑SOD, ↑GSH-Px, ↓p38 MAPK signal pathway, ↓SAPK/JNK pathway	[102]
				1, 5 and 10 μM				BV-2 cells		↓NO, ↓PGE2, ↓JNK, ↓p38 MAPK	[103]
								Neuron/Glial Cells		↓CD68, ↓OX42, ↓IL-6, ↓gp130, ↑BDNF	[104]
20	Galangin	Flavonoid	*Alpinia officinarum Hance*	6.25–400 μM				Rat adult brains		↓AChE	[105]
21	β-Caryophyllene	Sesquiterpene	*Piper nigrum* L.		16, 48, 144 mg/kg				APP/PS1 mice	↓COX-2, ↓TNF-α, ↓IL-1β	[106]
22	Piperine	Alkaloid		2.5, 5, 10, 25µM		Donepezil		SH-SY5Y cells		↓AChE	[107]
					5 mg/kg	Donepezil	Saline		Wistar rats	↓MDA, ↓NO	[108]
					5, 10 and 20 mg/kg	Donepezil hydrochloride			Wistar rats	↓lipid peroxidation, ↓acetylcholinesterase	[109]
23	Curcumin	Polyphenol	*Curcuma longa* L.		150 mg/kg				APP/PS1 mice	↓NF-κB pathway, ↑PPAR-γ	[110]
					50 mg/kg		PBS		5xFAD mouse	↓Aβ, ↓GFAP-IR, ↓Iba-1-IR	[111]
24	Aromatic-turmerone	Flavonoid		5, 10 and 20 μM				HT-22 cells		↓MMP-9, ↓iNOS, ↓COX-2, ↓TNF-α, ↓IL-1β, ↓IL-6, ↓MCP-1, ↓ROS, ↓IκB-α, ↓JNK, ↓p38 MAPK.	[112]
				5, 10 and 20 μM	50, 100 mg/kg		PBS	BV2 cells	C57 mice	↓TNF-α, ↓IL-1β, ↓Myd88, ↓MAPK, ↓NF-κB	[113]
				50 µg/mL				Cerebellar granule neuron		↓cleaved caspase-3	[114]
25	Puerarin	Isoflavone	*Puerarialobata* (*Willd.*) *Ohwi*		80 mg/kg		Saline		Sprague–Dawley rats	↓tau hyperphosphorylation, ↓GSK-3β, ↓FGF-2	[115]
					30 mg/kg				APP/PS1 transgenic mice	↑HO-1, GSK-3β/Akt signaling pathways	[116]
				100 μM				PC12 cells		↑Bcl-2, ↑p-Bad, ↓caspase-3, ↑Akt	[117]
26	Spinosin	C-glycoside flavonoid	*Ziziphi Spinosae Semen*		1.25, 2.5, 5 and 10 mg/kg		0.5% CMC		ICR mice	↑ERK–CREB–BDNF signaling	[118]
				6.25, 12.5, 25 μM				Neuro-2a cells		↓APP, ↓BACE1, ↑ADAM10, ↓ROS, ↑Nrf2, ↑HO-1, Nrf2/HO-1 signaling pathway	[119]
					10 and 100 µg/kg				Kunming mice	↓MDA, ↓Aβ1–42, ↑BDNF, ↑Bcl-2, ↓IL-6	[120]
				30 μM		Donepezil		5XFAD hippocampus		↑LTP, ↑plasmin	[121]
				0.1, 1, and 10 μg/mL	4, 20, 100 and 500 μg/mL	Donepezil		Hippocampus	ICR mice	↑choline acetyltransferase	[122]
27	Liquiritin	Flavonoids	*Glycyrrhiza uralensis Fisch.*		25, 50, 100 mg/kg	Donepezil (3 mg/kg)			SD rats	↑GSH-Px, ↑SOD, ↓8-OHdG, ↓MDA	[123]
28	Glabridin	Flavonoid			1, 2 and 4 mg kg^–1^	Scopolamine (0.5 mg kg^–1^)	0.3 % carboxymethyl cellulose suspension		Kunming mice	↓cholinesterase	[124]
29	Liquiritigenin	Flavonoid		2.5, 5, 10, 20, 50, and 100 μM	30 mg/kg		0.1% sodium carboxymethyl cellulose	N2A cells; BV2 cells	APP/PS1 double transgenic mice	↓NLRP3, ↓cleaved caspase-1, ↓caspase-3, ↓IL-1β, ↓TNF-α, ↑IL-4, ↑IL-13, ↓BAX, ↑Bcl-2, ↓Aβ, ↑NEP, ↑IDE	[125]
30	Sulforaphene	Isothiocyanates	*Raphanus sativus* L.	0.5, 1, 2, 4, 8, 16, and 32 μM	25, 50 mg/kg	Donepezil (5 mg/kg)	ddH_2_O	BV-2 cells	SD rats	↓TNF-α; IL-6; ↑IL-10, ↑p-Akt (Ser473)/Akt, ↑p-GSK-3β (Ser9)/GSK-3β, ↓NO, ↓NF-κB	[126]
31	Gastrodin	Phenolic Glycosides	*Gastrodia elata Blume*		60 mg/kg		Saline		Sprague–Dawley rats	↓Aβ1-40/42, ↓APP, ↓β-site APP-cleaving, ↓Beclin-1, ↓LC3-II, ↓p62, ↓Bax, ↑Bcl-2, P38 MAPK signaling pathway	[127]
32	P-coumaric acid	Phenylacrylic acid	*Cornus officinalis Siebold & Zucc.*	5, 25, 50 µM				PC12 cells		↓BACE1, ↓iNOS, ↓p65, ↓ERK1/2, ↓JNK	[128]
33	Ursolic acid	Pentacyclic triterpenoids		1,10, 20 µM				PC12 cells		↓iNOS, ↓COX-2, ↓p65, ↓p-38, ↓JNK, ↓ERK	[128]

## Data Availability

Not applicable.

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
