# Peer review of "Medicine-Food Herbs against Alzheimer’s Disease: A Review of Their Traditional Functional Features, Substance Basis, Clinical Practices and Mechanisms of Action"

_molecules, 2022, doi:10.3390/molecules27030901_

Round 1

Reviewer 1 Report

This is a very interesting and comprehensive review on herbs against Alzheimer's disease. While this topic is of significance and of importance to compile information in such a review, it would benefit from the following points.:

1. There are some tables but no figures. Certain figures should be added for visual explanation of some topics i.e., structures of some active gradients in these herbal drugs, mechanism of action of some of them wherever known.

2. There are a lot of references however some of the important ones in the field are missing, for example:

-β-Cyclodextrin and Curcumin, a Potent Cocktail for Disaggregating and/or Inhibiting Amyloids: A Case Study with α-Synuclein Biochemistry 2014, 53, 25, 4081–4083

-Curcumin modulates α-synuclein aggregation and toxicity (2013ACS Chem. Neurosci. 4393 407

3. While the title and abstract of the review focuses on general herbal medicines (one would presume around the world) however the focus of the review is only on Chinese medicinal herbs. It should be noted that several of them are being used in other traditional/herbal medicine around the world. References and credit to these other/alternate usage in other geographical areas should be made.

Author Response

Manuscript ID: molecules-1506547 entitled " Medicine-food herbs against Alzheimer's disease: A review of their traditional functional features, substance basis, clinical practices and mechanisms of action" had been revised in accordance with the comments of the reviewers. Please find following our response to the comments made by the reviewers. Thank you for consideration of publication of this manuscript in Molecules.

Thank you for kind attention!

Respectfully,

Ailin Liu

Institute of Materia Medica, Chinese Academy of Medical Sciences and

Peking Union Medical College

A2 Nanwei Road, Xicheng District

Beijing 100050, China

Response to Reviewer 1 Comments

comment 1: There are some tables but no figures. Certain figures should be added for visual explanation of some topics i.e., structures of some active gradients in these herbal drugs, mechanism of action of some of them wherever known.

Response: We sincerely thank the reviewer for pointing out deficiencies and content for improving the quality of this manuscript. We carefully considered the content of the article, and visualized the active components of MFH (Figure 1-Figure 3) and its known mechanism (Figure4-Figure5) in the MS according to your valuable opinions. The main active ingredients of MFH included in this paper are saponins and flavonoids. In addition, there are phenylethanoid, aromatic acid, aromatic ester, furfural and olefinic alcohol, etc.(Please see the attachment)

Figure 1. The ingredients of Saponin in MFH.

Figure 2. The ingredients of Flavonoid in MFH.

Figure 3. Other active ingredients in MFH.

Figure 4. MFH anti-AD network mechanism diagram. (Blue nodes represent MFH, red nodes represent targets/pathways)

Figure 5. The ingredient of MFH anti-AD network mechanism diagram. (Blue nodes represent ingredient, red nodes represent targets/pathways)

Comment 2: There are a lot of references however some of the important ones in the field are missing, for example:

-β-Cyclodextrin and Curcumin, a Potent Cocktail for Disaggregating and/or Inhibiting Amyloids: A Case Study with α-Synuclein Biochemistry 2014, 53, 25, 4081–4083

-Curcumin modulates α-synuclein aggregation and toxicity (2013) ACS Chem. Neurosci. 4, 393– 407

Response: We are very sorry for the omission of some information in the references. We have added the references according to the Reviewer’s comments.

Comment 3: While the title and abstract of the review focuses on general herbal medicines (one would presume around the world) however the focus of the review is only on Chinese medicinal herbs. It should be noted that several of them are being used in other traditional/herbal medicine around the world. References and credit to these other/alternate usage in other geographical areas should be made.

Response: Thanks for your professional comment. First of all, we are sorry for the mismatch between the title and the content in the manuscript. According with your advice, we amended the relevant part in manuscript. The title is changed to " Medicine-food herbs in Chinese traditional medicines against Alzheimer's disease: A review of the traditional functional features, substance basis, clinical practices and mechanisms of action ".

In this study, we only focus on the related review of homology of medicine and food in traditional Chinese medicines. To ensure the safety of functional foods from TCM, the National Health Commission of the People's Republic of China released specific provisions on MFH items, which considered 109 TCMs as functional foods for alleviating or curing various chronic diseases through the diet as of 2019 (Hou & Jiang, 2013).

Reference:

Hou, Y., & Jiang, J. G. (2013). Origin and concept of medicine food homology and its application in modern functional foods. Food Funct, 4(12), 1727-1741. doi:10.1039/c3fo60295h

Reviewer 2 Report

In this manuscript the authors have summarised the different traditional Chinese medicines and have compiled clinical evidence of the same with regards to the alleviation of the neurodegenerative effects of amyloid beta (Ab) aggregation. Extensive research is being carried to counter the degenerative effects of such aggregates. Traditional medicines that are generally included in diet for certain other ailments are increasingly being shown to be effective without having any severe side effects. Thus, the present review that highlights all the important traditional medicines, their source and their mechanisms of action (of the components) comes at a very opportune time which other researchers can use as a ready reference instead of having to go through an extensive literature search. The tables are well made with all possible details including recommended dosages and the relevant clinical trials along with the associated findings. The MOA details are going to be very informative in the worldwide efforts to further the goal towards addressing AD and its symptoms using the traditional medicines that have huge potential. The individual components have been reported to possess a wide range of health benefits. The fact that these are also useful in their own ways to alleviate symptoms related to Ab aggregation suggest that different ailments might have underlying links with the development of Ab aggregation induced pathogenesis. Tables 2 and 3 are the most important ones and will be finding widespread usage. Going through the review one will feel that a possible way of combating the neurodegenerative disease from different aspects is by having a concoction of selected compounds reported and maybe studies related to this can be carried out in future if not already being attempted in different research labs.

Recommendation: Should be published after minor revisions

Other Comments:

  • Please make sure the Table numbers are sequential.

Author Response

Manuscript ID: molecules-1506547 entitled " Medicine-food herbs against Alzheimer's disease: A review of their traditional functional features, substance basis, clinical practices and mechanisms of action" had been revised in accordance with the comments of the reviewers. Please find following our response to the comments made by the reviewers. Thank you for consideration of publication of this manuscript in Molecules.

Thank you for kind attention!

Respectfully,

Ailin Liu

Institute of Materia Medica, Chinese Academy of Medical Sciences and

Peking Union Medical College

A2 Nanwei Road, Xicheng District

Beijing 100050, China

Response to Reviewer 2 Comments

Comment 1: In this manuscript the authors have summarised the different traditional Chinese medicines and have compiled clinical evidence of the same with regards to the alleviation of the neurodegenerative effects of amyloid beta (Ab) aggregation. Extensive research is being carried to counter the degenerative effects of such aggregates. Traditional medicines that are generally included in diet for certain other ailments are increasingly being shown to be effective without having any severe side effects. Thus, the present review that highlights all the important traditional medicines, their source and their mechanisms of action (of the components) comes at a very opportune time which other researchers can use as a ready reference instead of having to go through an extensive literature search. The tables are well made with all possible details including recommended dosages and the relevant clinical trials along with the associated findings. The MOA details are going to be very informative in the worldwide efforts to further the goal towards addressing AD and its symptoms using the traditional medicines that have huge potential. The individual components have been reported to possess a wide range of health benefits. The fact that these are also useful in their own ways to alleviate symptoms related to Ab aggregation suggest that different ailments might have underlying links with the development of Ab aggregation induced pathogenesis. Tables 2 and 3 are the most important ones and will be finding widespread usage. Going through the review one will feel that a possible way of combating the neurodegenerative disease from different aspects is by having a concoction of selected compounds reported and maybe studies related to this can be carried out in future if not already being attempted in different research labs.

Response : Thank you for your professional comment and your high recognition of our work.

Comment 2: Please make sure the Table numbers are sequential.

Response : Thanks for your valuable comment. We have carefully corrected the Table numbers in the manuscript according to your opinion.

Round 2

Reviewer 1 Report

After the comments addressed by the authors, I feel that the manuscript can be published.